https://doi.org/10.5194/egusphere-2025-4460 Preprint. Discussion started: 18 November 2025 © Author(s) 2025. CC BY 4.0 License.

- Measurement report: Nitrogen Isotope (δ<sup>15</sup>N) Signatures of Ammonia Emissions
- from Livestock Farming: Implications for Source Apportionment of Haze
- Pollution
- Jinhan Wang<sup>1</sup>, Zhaojun Nie<sup>1</sup>, Yupeng Zhang<sup>1</sup>, Xiaolei Jie<sup>1,2</sup>, Haiyang Liu<sup>1</sup>, Peng
- Zhao<sup>1,2,3</sup>, Hongen Liu<sup>1,2,3</sup>
- College of Resources and Environment, Henan Agricultural University, Zhengzhou, Henan 450046,
- China
- <sup>2</sup> Key Laboratory of Farmland Quality Conservation in the Huang-Huai-Hai Plain, Ministry of
- Agriculture and Rural Affairs, Zhengzhou 450046, China;
- 3 Key Laboratory of Soil Pollution Prevention, Control and Remediation in Henan Province, Zhengzhou
- 450046, China.
- Correspondence to: Hongen Liu, Email: liuhongen7178@126.com; Yupeng Zhang, Email:
- zhangyp@henau.edu.cn
- **Abstract.** Ammonia emissions from agriculture are the primary source of atmospheric reactive nitrogen,
- significantly impacting air pollution, soil acidification, eutrophication of water bodies, and human health.
- Accurate quantification of ammonia from different sources is crucial for effective mitigation. In this
- study, the air extraction method was employed to collect gases from livestock farms, and the δ<sup>15</sup>N values
- $18 \qquad \text{of volatilized ammonia (NH$_3$) from the animal husbandry industry in the southern Huang Huai Hai} \\$
- 19 Plain of China were analyzed using stable nitrogen isotopes. The results show that isotopic signatures
- differ significantly among livestock types: dairy cows (-20.6%  $\pm$  0.8%), laying hens (-27.4%  $\pm$  1.0%),
- and pigs ( $-38.4\% \pm 1.7\%$ ). These livestock-derived signatures are distinct from those associated with
- 22 combustion sources (-7.0%  $\pm$  2.1%) and traffic emissions (6.6%  $\pm$  2.1%), and they exhibit considerably
- 23 lower variability than fertilizer-derived signatures. Overall, this work provides high-precision isotopic
- 24 source signatures for livestock operations, offering essential parameters for regional atmospheric
- ammonia source apportionment and highlighting the need for locally tailored mitigation strategies.

27 Graphical Abstract.

26

28

29

30

31

32

33

34

35

36

37

38

39

40

41

47

#### 1. Introduction

Ammonia (NH<sub>3</sub>) is a highly reactive and abundant nitrogenous gas in the atmosphere. It is classified as a major alkaline species and readily reacts with sulfuric acid and nitric acid to produce ammonium sulfate ((NH<sub>4</sub>)<sub>2</sub>SO<sub>4</sub>) and ammonium nitrate (NH<sub>4</sub>NO<sub>3</sub>) (Kawashima et al., 2023; Kirkby et al., 2011). These compounds can form particulate ammonium salts or interact with organic aerosols to generate secondary aerosols. In moderately polluted environments, the mass fraction of these ammoniumcontaining particles within PM<sub>2.5</sub> is relatively low (Huang et al., 2014; Yang et al., 2011). Under severe pollution conditions, however, ammonium sulfate, ammonium nitrate, and other ammonium salts can account for up to approximately 50% of the total PM<sub>2.5</sub> mass (Battye, 2003; Beusen et al., 2008; Goebes et al., 2003). As a key precursor of secondary inorganic aerosols, NH3 is a primary contributor to haze formation and constitutes a substantial component of PM<sub>2.5</sub> in polluted atmospheres (Wu et al., 2024; Xiang et al., 2022). Excessive ammonia emissions also drive a range of environmental problems, including soil acidification, climate perturbation, reduced atmospheric visibility, and eutrophication of aquatic ecosystems (Huang et al., 2012; Jiang et al., 2021). Consequently, reducing NH<sub>3</sub> emissions has recently been proposed as a strategy to mitigate smog pollution in China (Liu et al., 2019). Over the past few decades, substantial changes in air quality have been observed across many countries worldwide (Boyle, 2017; Warner et al., 2017). Notably, China has consistently ranked first in global ammonia (NH<sub>3</sub>) emissions (Liu et al., 2013). Current NH<sub>3</sub> emission inventories identify the principal sources as agricultural activities-including fertilizer application and livestock and poultry farming-and non-agricultural sources, such as combustion processes and vehicular emissions (Bouwman

56

et al., 1997; Schlesinger and Hartley, 1992; Streets et al., 2003). It is widely recognized that agriculture represents the predominant source of atmospheric NH<sub>3</sub>, contributing over 70% of total emissions (Meng et al., 2017; Xu et al., 2024), accounting for more than 70% of the total (Ma et al., 2021; Ti et al., 2019), with livestock and poultry farming alone accounting for for 50% to 60% of agricultural NH<sub>3</sub> emission (Huang et al., 2012; Wang et al., 2018). Despite this, substantial uncertainty remains regarding the contribution of livestock-derived NH<sub>3</sub> to nitrogen deposition (Elliott et al., 2019), and estimating these contributions using satellite remote sensing and livestock emission inventories remains challenging (Beusen et al., 2008; Li et al., 2023a; Van Damme et al., 2018). These conventional approaches typically rely on fixed emission factors, such as unit animal excretion coefficients, which are limited by temporal lags and insufficient spatial resolution, thereby hindering the capture of real-time variations in NH<sub>3</sub> emissions and the resulting nitrogen deposition at the farm scale. In contrast, nitrogen stable isotope analysis ( $\delta^{15}$ N) provides a direct and highly effective approach for tracing the sources of NH<sub>3</sub> and NH<sub>4</sub><sup>+</sup> (Bhattarai et al., 2020; Xiao et al., 2020). This methodology relies on the principle that distinct emission sources and environmental processes generally exhibit unique isotopic fingerprints (Elliott et al., 2019; Li et al., 2024; Sui et al., 2020), defined by the ratio of heavy (15N) to light (14N) nitrogen isotopes in collected samples (Song et al., 2021). Numerous studies have employed stable nitrogen isotope ( $\delta^{15}$ N) techniques to quantify the contributions of combustion, transportation, and agricultural activities to atmospheric NH3 and NH4+ (Xiang et al., 2022; Xie et al., 2008). For example, during the corn growing season in Northeast China, δ<sup>15</sup>N values of NH<sub>3</sub> volatilized from farmland exhibited a wide range, from -38.0% to -0.2%. Notably,  $\delta^{15}$ N emission rates were considerably lower during the early stages of corn growth compared to later stages, indicating clear seasonal variation (Song et al., 2024). Under different fertilization regimes, significant differences in  $\delta^{15}$ N-NH<sub>3</sub> emissions were observed, with values fluctuating between -46.0% and -4.7% throughout the volatilization period (Ti et al., 2021). Previous studies report that  $\delta^{15}$ N-NH<sub>3</sub> and  $\delta^{15}\text{N-NH}_4^+$  emissions from combustion sources (-7.6‰ to +16.2‰) predominate in winter, contributing up to 51.6% of total ammonia emissions (Xiao et al., 2022, 2025; Zhou et al., 2021). In contrast, NH<sub>3</sub> emissions from vehicle exhaust exhibit relatively high  $\delta^{15}$ N values (13.7 ± 3.7%) (Savard et al., 2017; Xi et al., 2023). However, these emissions are primarily localized in urban environments. Currently, limited studies have reported the  $\delta^{15}N$  characteristics of ammonia from livestock and poultry farming. Existing data mostly rely on passive sampling methods (Berner and David Felix, 2020;

surrounding farms (pig farms: -35.1% to -10.5%; cattle farms: -24.7% to -11.3%). Additional research 80 has quantified  $\delta^{15}$ N variability in livestock and poultry (-31.0% to -15.0%) through simulated ammonia 81 emissions during manure management processes (Hristov et al., 2009). It is noteworthy that  $\delta^{15}$ N-NH<sub>3</sub> 82 fluctuations in livestock and poultry operations may also depend on animal growth stages and 83 reproductive status.. 84 The MixSAIR model has primarily been employed to apportion the contributions of atmospheric emission sources using isotope analysis (Chang et al., 2016; Walters et al., 2022). However, there is no 85 86 universally fixed  $\delta^{15}$ N-NH<sub>4</sub><sup>+</sup> value for each emission source. As a result, substantial variations in reported 87  $\delta^{15}\text{N-NH}_4^+$  values for the same source have been documented across different studies. To date, no 88 research has validated changes in  $\delta^{15}$ N-NH<sub>4</sub><sup>+</sup> resulting specifically from livestock and poultry farm 89 emissions, nor has the relationship between  $\delta^{15}$ N-NH<sub>4</sub><sup>+</sup> from different sources and regional variations 90 been examined. To obtain more accurate assessments of  $\delta^{15}$ N-NH<sub>3</sub> variations associated with ammonia 91 emissions from livestock and poultry farming, and to achieve reliable atmospheric NH<sub>3</sub> source 92 apportionment, it is essential to characterize the correlation between  $\delta^{15}$ N-NH<sub>4</sub><sup>+</sup> from different sources 93 and regional differences. In this study, active dynamic sampling methods were used to collect ammonia 94 emissions from intensive pig farms, dairy farms, and laying hen farms located in the southern region of 95 the Huang-Huai-Hai Plain. Meta-analysis techniques were employed to analyze the  $\delta^{15}$ N signatures of 96 different ammonia emission sources. The specific objectives of this research are: (1) to determine the 97  $\delta^{15}$ N-NH<sub>4</sub><sup>+</sup> values of emissions from livestock and poultry housing at various growth stages; and (2) to 98 investigate the relationship between  $\delta^{15}$ N-NH<sub>4</sub><sup>+</sup> from different sources and regional variations.

Chang et al., 2016; Ti et al., 2018), which assess  $\delta^{15}N$  changes by collecting wet deposition samples

### 2. Materials and methods

## 2.1. Sampling points in the study area and sample collection and processing.

The sampling experiment at the farm was conducted from May 9, 2024, to December 6, 2024. No samples were collected in July and August due to the absence of livestock or poultry during these months. The collected samples covered the entire breeding period of fattening pigs and the period from chicks to peak egg production in laying hens. Throughout the trial period, six batches of samples were obtained, amounting to a total of 120 samples for measuring ammonia emissions from livestock and poultry

as severe, whereas samples collected under clean atmospheric conditions corresponded to air quality classified as excellent. Samples were collected using atmospheric samplers (Beijing Ke'an Labor Protection Company) at a flow rate of 0.1 to 2 L·min<sup>-1</sup>, with each sample collected over a duration of 60 minutes. The intensive fattening pig farm is located in Luoyang City, Henan Province (112.71° E, 34.52° N), with no other livestock operations in the surrounding area. The sampled fattening pig farm houses 2,600 pigs distributed across four fully enclosed pig houses. One of these houses was selected as the target sampling site. The sampling procedure was as follows: an atmospheric sampler was positioned 2.0 meters from the exhaust vent of the livestock and poultry house at a height of 1.6 meters, corresponding to the central height of the exhaust outlet. T The sampling duration was set to 60 minutes, with the gas flow rate maintained at 2 L·min<sup>-1</sup> using a flow meter. A bubbler absorption bottle filled with absorption solution was used to collect NH3. Three atmospheric samplers were operated simultaneously during each sampling event. Figure 1 marks the sampling points of the intensive pig farms with green pentagrams. In the case of intensive laying hens farms, each building houses approximately 15,000 laying hens and is fully enclosed, with a total of 300,000 laying hens being raised. The sampling site is located in Zhengzhou City, Henan Province (114.03° E, 34.59° N). One building was selected as the target sampling point, with the sampling method mirroring that used for the fattening pig farms. As shown in Figure 1, the light blue pentagons represent the sampling points of intensive layer farms. The intensive dairy farm operates with an open-style barn design, housing 400 dairy cows per barn, with a total of 4,000 dairy cows being raised. Four atmospheric samplers were installed in the passageways of the dairy barns, with each sampler spaced 10 meters apart and positioned at a height of 1.6 meters. The dairy farm is located in Zhengzhou City, Henan Province (114.11° E, 34.81° N). The sampling time and method remained consistent with those described above. In Figure 1, the dark blue pentagons represent the sampling points of intensive dairy farms. To investigate the variations in  $\delta^{15}$ N levels associated with differing degrees of air pollution, samples collected for  $\delta^{15}N$  measurement during periods of severe smog and when air quality was pristine. The sampling location was situated on a spacious lawn within the campus of Henan Agricultural University, devoid of tall buildings or traffic. The sampling point is illustrated in Figure 1, where the pink triangle represents the sampling site for both haze and clean air (Longitude 113.82° E, Latitude 34.80° N). Each

housing. On days when samples were collected during hazy weather, the air pollution level was classified

156157

158

- sampling event utilized three atmospheric samplers, positioned at a height of 1.6 meters, with the duration
   of sampling aligned with that of the livestock farm.
- The collected sample solution is transferred into a centrifuge tube and returned to the laboratory,
  where the concentration of NH<sub>3</sub> is measured using a UV spectrophotometer. The detection method
  adheres to the guidelines outlined in "Determination of Ammonia Nitrogen in Water by Salicylic Acid
  Spectrophotometry" (HJ 536-2009), and the calculation method is presented in Equation (1):

$$\rho_N = \frac{A_s - A_b - a}{b \times V} \times D \tag{1}$$

- Where,  $\rho_N$  represents the mass concentration of ammonia nitrogen in the water sample (expressed as N), 144 in mg·L<sup>-3</sup>. The variables are defined as follows:  $A_s$  denotes the absorbance of the sample, while  $A_b$ 145 indicates the absorbance of the blank experiment, which is prepared from the same batch as the sample. 146 The parameters a and b correspond to the intercept and slope of the calibration curve, respectively. 147 Additionally, V refers to the volume of the water sample taken, measured in mL, and D signifies the 148 dilution factor of the water sample.
  - The analytical method described employs the bromate-hydroxylamine chemical approach (Soler-Jofra et al., 2016; Zhang et al., 2007). Initially, a potassium bromate-potassium bromide solution reacts under acidic conditions to produce bromine, which subsequently reacts in a strongly alkaline environment to generate bromate, a potent oxidizing agent capable of oxidizing NH<sub>4</sub><sup>+</sup> to NO<sub>2</sub><sup>-</sup>. In the following step, hydroxylamine hydrochloride reduces NO<sub>2</sub><sup>-</sup> in an acidic environment to form N<sub>2</sub>O. The resultant N<sub>2</sub>O is then analyzed using a stable isotope ratio mass spectrometer, along with a multi-purpose online gas preparation device, and an automatic sampler, to determine the  $\delta^{15}$ N value. For each sample analysis, four international standard materials for NH<sub>4</sub><sup>+</sup> (IAEA-N-1, USGS-25, IAEA-N-2, and USGS-26, with  $\delta^{15}$ N concentrations of 0.4‰, -30.41‰, 20.3‰, and 53.75‰, respectively) are processed simultaneously.

Figure 1. Sampling sites of livestock farms, haze weather, and clear weather in this study, extracted from the main research sampling locations. Yellow dots represent the main global research sampling sites, pink triangles represent sampling sites during haze and clear weather, dark blue pentagons represent cattle farms, light blue pentagons represent layer farms, and green pentagons represent fattening pig farms.

### 2.2. Data collection and processing

We screened articles published between January 2000 and January 2025 regarding the sources of  $\delta^{15}\text{N-NH}_3$  and  $\delta^{15}\text{N-NH}_4^+$ . Specifically, we utilized ISI Web of Science, Google Scholar, and PubMed, employing the search terms " $\delta^{15}\text{N}$ ," "NH<sub>3</sub>," "ammonia emissions," and "isotopes" to identify relevant literature. Studies included in our analysis were required to meet the following criteria: (1) Samples must be measured for either  $\delta^{15}\text{N-NH}_3$  or  $\delta^{15}\text{N-NH}_4^+$ ; (2) Experiments must encompass at least one of the following: combustion, fertilization, agriculture, transportation, or livestock farming; (3) The number of experimental replicates and sampling events must be explicitly reported; (4) Samples must primarily consist of atmospheric NH<sub>3</sub> or PM<sub>2.5</sub>, and detection must employ chemical methods. A total of 37 documents were included in the analysis. This dataset comprehensively encompasses multiple meta-analyses and original studies, detailing changes in  $\delta^{15}\text{N-NH}_3$  and  $\delta^{15}\text{N-NH}_4^+$  from combustion sources, transportation sources, agricultural sources, and livestock farming sources; the proportion of  $\delta^{15}\text{N}$  values

185186

196197

in the atmosphere; geographical location (latitude and longitude); and the GDP of each city where samples were collected. If the data in the literature was presented solely in chart form, we utilized WebPlotDigitizer-4.7 (https://apps.automeris.io/wpd4/) to extract the data. We categorized the collected data into five distinct groups: combustion, transportation, farmland, livestock farming, and PM<sub>2.5</sub>.

A total of 126 samples were collected, and 41 literature references were gathered. Data analysis was performed using Excel, SPSS, and Python version 3.11.

### 3. Result and discussion

#### 3.1. Temporal Variations in Ammonia Emissions and $\delta^{15}N$ Signatures from Livestock Farms

During the sampling period from May to December, ammonia emissions varied significantly among the three farm types: 4.9 to 6.7 mg·m<sup>-3</sup> for fattening pigs (Figure 2a), 1.7 and 2.5 mg·m<sup>-3</sup> for dairy cows (Figure 2b), and 3.8 to 7.1 mg·m<sup>-3</sup> for laying hens (Figure 2c), with the latter exhibiting substantial temporal fluctuations. NH<sub>3</sub> emissions from fattening pigs peaked when the pigs reached 130 kg·head<sup>-1</sup> (Figure 2a), For laying hens, NH3 concentrations initially increased and subsequently declined in response to temperature variations, reflecting enhanced urease activity within the housing environment, which accelerates urea hydrolysis and promotes NH<sub>3</sub> volatilization.δ<sup>15</sup>N-NH<sub>4</sub><sup>+</sup> levels at the livestock farms showed significant temporal variation (p < 0.05) (Groot Koerkamp et al., 1998; Rosa et al., 2020). From May to June, the  $\delta^{15}$ N-NH<sub>4</sub><sup>+</sup> From May to June,  $\delta$ 15N-NH4+ increased from -31.0% to -25.2% in fattening pig farms and from -26.4% to -24.6% in laying hen farms. In September,  $\delta^{15}$ N-NH<sub>4</sub>+ values from fattening pig farms  $(-13.3 \pm 1.3\%)$  were significantly higher than those from laying hen and dairy cow farms (-13.9  $\pm$  0.9%), which were comparable. Over the following three months,  $\delta^{15}$ N-NH<sub>4</sub><sup>+</sup> levels decreased significantly across both farm types. As illustrated in Figure 2, the highest NH<sub>3</sub> concentration at the dairy farm  $(2.5 \pm 0.3 \text{ mg} \cdot \text{m}^{-3})$  occurred in October, coinciding with the lowest  $\delta^{15}\text{N-NH}_4^+$  values. while laying hen farms also recorded minimum δ<sup>15</sup>N-NH<sub>4</sub>+ during this period of elevated NH<sub>3</sub>. Conversely, the lowest δ<sup>15</sup>N-NH<sub>4</sub><sup>+</sup> at fattening pig farms was observed in December, despite peak NH<sub>3</sub> concentrations. NH<sub>3</sub> concentrations differed significantly between hazy and clear weather in December (Figure 2d), with  $\delta^{15}$ N-NH<sub>4</sub><sup>+</sup> values being significantly higher under clear conditions (1.9  $\pm$  0.8‰) than under hazy conditions (1.6  $\pm$  0.2%; p 

Figure 2. Changes in NH<sub>3</sub> emissions and  $\delta^{15}$ N-NH<sub>4</sub><sup>+</sup> values outside the livestock farms among different months. (a)Fatting pig farm; (b)Dairy cow farm; (c) Laying hens farm; (d) Comparison of Haze and clean air samples. Statistical difference was calculated by T-test, P < 0.05, n = 3.

As illustrated in Figure 3, throughout the entire monitoring period, ammonia (NH<sub>3</sub>) sources form the farms exhibited nitrogen depletion, indicated by negative  $\delta^{15}\text{N-NH}_4^+$  values. Overall,  $\delta^{15}\text{N-NH}_4^+$  values exhibited significant fluctuations in dairy and fattening pig farms, while variations were comparatively moderate in laying hens farms. Notably, the  $\delta^{15}\text{N-NH}_4^+$  values at dairy cattle farms displayed substantially greater overall changes during the monitoring period compared to those in laying hens and fattening pig farms. The arithmetic mean value at fattening pig farms was -30.8  $\pm$  1.6‰, the lowest among the three types of farms, whereas the  $\delta^{15}\text{N-NH}_4^+$  values in laying hens manure remained at an intermediate level throughout the entire period. From October to December, the  $\delta^{15}\text{N-NH}_4^+$  values at livestock and poultry farms were generally lower than those observed in the first half of the monitoring period (Figure 3). However, when comparing hazy and clear weather conditions, the  $\delta^{15}\text{N-NH}_4^+$  values for all three types of farms consistently remained at a relatively low level during this timeframe (Figure 3).

Figure 3. Changes of  $\delta^{15}$ N-NH<sub>4</sub><sup>+</sup> abandance at intensive livestock farms during the sampling period. Hazy and clean air were also sampled at December. The air sample of laying hens in December was missed, because of death of chicken by avian influenza.

## 3.2. Comparison with Literature and Implications for Local Sources

During the monitoring period, the  $\delta^{15}$ N-NH<sub>4</sub><sup>+</sup> values ranged from -50.0% to -10.0% (Figure 4a). For fattening pigs,  $\delta^{15}$ N-NH<sub>4</sub><sup>+</sup> values averaged -38.4%  $\pm$  1.8% between October and December, which was significantly lower than the previously reported range of -27.10% to -31.7% (Chang et al., 2016) Notably, the overall variation remained within the  $\delta^{15}$ N-NH<sub>4</sub><sup>+</sup> emission ranges report for fattening pigs in other studies (Bhattarai and Wang, 2023; Wang et al., 2022). Furthermore, due to differences in livestock management practices and nitrogen content in feed, the  $\delta^{15}\text{N-NH}_4^+$  values from dairy farms in this study, averaging -29.4% ± 13.9%, were substantially lower than those reported by Martine M et al.  $(20.5\% \pm 34.5\%)$  (Savard et al., 2017). Comparison with  $\delta^{15}$ N-NH<sub>4</sub><sup>+</sup> values measured in dairy farms in Akita, Japan, were -22.5%  $\pm$  -14.6% (Kawashima, 2019), no significant difference was observed relative to the values obtained in this study. https://doi.org/10.5194/egusphere-2025-4460 Preprint. Discussion started: 18 November 2025 © Author(s) 2025. CC BY 4.0 License.

244245

247248

37.9% to -22.9% based on passive sampling techniques. Previous research has shown that active sampling generally yields higher  $\delta^{15}$ N values than passive sampling (Kawashima and Ono, 2019; Pan et al., 2020). This discrepancy arises from the diffusion-driven nature of passive samplers, in which lighter NH<sub>3</sub> molecules are preferentially adsorbed. Consequently, passive sampling typically produces  $\delta^{15}$ N values that deviate by approximately 15% from those obtained by active sampling (Bhattarai and Wang, 2023; Skinner et al., 2006). Variations in  $\delta^{15}$ N-NH<sub>4</sub><sup>+</sup> values are known to occur among different livestock species. During the monitoring period,  $\delta^{15}N-NH_4^+$  values from laying hen farms were consistently lower than those from dairy farms but higher than those from fattening pig farms, consistent with previously reported trends. This pattern suggests that  $\delta^{15}$ N-NH<sub>4</sub><sup>+</sup> variations in emitted NH<sub>3</sub> are not primarily driven by animal body weight but are instead strongly modulated by environmental conditions (Choi et al., 2017; Qu and Zhang, 2021). In agreement with earlier studies, δ<sup>15</sup>N-NH<sub>4</sub>+ emissions from fattening pig and laying hen farms differed significantly from previously documented values, whereas no significant difference was observed for dairy cattle farms. Furthermore, the magnitude of  $\delta^{15}N-NH_4^+$  fluctuations across the three farm types was smaller than that reported in earlier literature. Comparison with major atmospheric NH<sub>3</sub> sources further demonstrated that the  $\delta^{15}$ N-NH<sub>4</sub> values measured in this study diverged substantially from those associated with combustion (-7.0%  $\pm$  2.1%), fertilization application (-38.0%)  $\pm$  0.2‰), and transportation (6.6‰  $\pm$  2.1‰). Based on  $\delta^{15}$ N-NH<sub>4</sub><sup>+</sup> signatures measured under both hazy and clear weather conditions, it can therefore be inferred that agricultural and livestock emissions are not the dominant contributors to atmospheric NH3 in Zhengzhou. Instead, traffic exhaust and combustion sources appear to constitute the primary contributors.

Figure 4. Comparison of  $\delta^{15}N$ -NH<sub>4</sub><sup>+</sup> values within different livestock farms and historical reported data. (a)Comparison of the  $\delta^{15}N$ -NH<sub>4</sub><sup>+</sup> values among different livestock farms; (b)Comparison of the  $\delta^{15}N$ -NH<sub>4</sub><sup>+</sup> values from present study with previously reported data.

# $\textbf{3.3. Global Variability of NH}_{\textbf{3}} \, \textbf{Source Signatures and Challenges for Source Apportionment}$

Ammonia emissions that contribute to urban smog primarily arise from combustion activities, vehicle exhaust, agriculture fertilization, and livestock production. As national economies expand, the frequency and severity of smog events have intensified. Figure 5a (slope: 0.026, intercept: 1.6323, R<sup>2</sup>:

287

0.0963) shows that from 2000 to 2025, when GDP remains below 70 billion USD, atmospheric  $\delta^{15}$ N-NH<sub>4</sub><sup>+</sup> signatures predominantly reflect fertilizer-derived emissions from agricultural regions and NH<sub>3</sub> volatilization from livestock operations (Kawashima et al., 2022; Kawashima and Kurahashi, 2011). This pattern indicates that lower-income regions rely heavily on agriculture and animal husbandry as the foundational components of their economies (Leng et al., 2018).

When GDP increases to between 80 billion and 300 billion USD, the contribution of combustionrelated and vehicular sources to δ<sup>15</sup>N-NH<sub>4</sub><sup>+</sup> becomes increasingly prominent. Notably, vehicle exhaust remains the dominant contributor within this GDP interval, suggesting that transportation serves as a key economic driver during mid-stage development. In densely populated and economically advanced cities, rapid vehicle growth further amplifies the influence of transportation-related  $\delta^{15}$ N-NH<sub>4</sub><sup>+</sup> signatures(Lim et al., 2022; Pan et al., 2018; Stratton et al., 2019). Throughout the entire dataset, vehicle exhaust and combustion together account for nearly 70% of ammonia emissions(Wu et al., 2019). Once GDP surpasses 300 billion USD,  $\delta^{15}$ N-NH<sub>4</sub><sup>+</sup> from combustion becomes the dominant atmospheric source, while the relative contribution from vehicle exhaust begins to decline and emissions from agricultural fertilization and livestock farming become negligible (Li et al., 2023b). It is important to note that sampling sites in the present study were located near power plants (Lim et al., 2019; Zou et al., 2022), whereas comparison data from previous studies were collected in urban cores. This spatial difference further supports the conclusion that in highly developed cities, shifts in economic structure lead to combustion sources emerging as the principal contributors to atmospheric NH3 under both hazy and clear meteorological conditions. As illustrated in Figure 5b, the proportion of  $\delta^{15}N-NH_4^+$  attributed to combustion and vehicular sources has increased over time. This temporal trend suggests that, with economic growth, agricultural and livestock emissions no longer represent the dominant contributors to atmospheric ammonia.

Figure 5. Changes of δ<sup>15</sup>N-NH<sub>4</sub><sup>+</sup> values among different GDP cities and years. (a) The relationship

between GDP and  $\delta^{15}N-NH_4^+$  values; (b) Changes of  $\delta^{15}N-NH_4^+$  values reported between 2008 to 2021.

The extracted dataset was classified into four major emission categories-livestock farming, combustion, farmland fertilization, and vehicle exhaust-and subsequently subjected to statistical evaluation. As illustrated in Figure 6,  $\delta^{15}$ N-NH<sub>4</sub>+ values associated with combustion sources showed strong consistency with previously reported ranges (Chang et al., 2021). Although traffic exhaust and livestock-related  $\delta^{15}$ N-NH<sub>4</sub>+ values exhibited moderate dispersion, both sources remained within relatively well-defined isotopic ranges. In sharp contrast,  $\delta^{15}$ N-NH<sub>4</sub>+ signatures following farmland fertilization displayed pronounced heterogeneity, covering nearly the entire isotopic spectrum reported for combustion, livestock, and vehicular emissions. This extensive variability highlights substantial regional differences in agricultural ammonia emission processes (Felix et al., 2014; Li et al., 2023b). Consequently, accurate source apportionment of atmospheric NH<sub>3</sub> requires distinguishing dominant local emission pathways rather than relying solely on generalized isotopic patterns (Chen et al., 2022; Zhang et al., 2023).

Figure 6. Statistical analysis of extracted data categorized by source: combustion sources, livestock and poultry farming sources, agricultural sources, and transportation exhaust sources.

## 4. Summary

This study establishes high-precision  $\delta^{15}N$  signatures for ammonia emissions from three dominant intensive livestock systems in the Huang-Huai-Hai Plain. Distinct isotopic fingerprints were identified

314315

for dairy operations (-20.6‰  $\pm$  0.8‰), laying hen facilities (-27.4‰  $\pm$  1.0‰), and fattening pig farms (-38.4‰  $\pm$  1.7‰), underscoring clear differences among livestock categories. Our results further demonstrate that isotopic signatures vary dynamically with NH<sub>3</sub> volatilization intensity, highlighting the need to incorporate volatilization-driven fractionation effects into isotope-based source apportionment frameworks. When compared with ambient  $\delta^{15}$ N-NH<sub>4</sub><sup>+</sup> measurements in Zhengzhou, the newly constrained source end-members indicate that non-agricultural sources-particularly vehicular emissions and combustion-are likely major contributors to urban atmospheric ammonia. This interpretation, however, requires validation through comprehensive isotopic mixing and dispersion modeling. Moreover, global-scale evaluation reveals that the exceptional variability of  $\delta^{15}$ N associated with fertilized soils continues to pose a substantial challenge for accurate identification of agricultural contributions. Collectively, the findings presented here provide critical isotopic constraints that can enhance regional atmospheric chemistry models and support the design of more precise and effective ammonia emission control policies.

## **Author Contributions**

- 321 J.W. Drafting, Formal Analysis, Data Management, Methodology, Investigation; Z.N. Formal Analysis,
- Data Management, Methodology, Investigation; Y.Z. Conceptualization, Data Management,
- Visualization, Funding Acquisition, Drafting, Formal Analysis, Writing Review & Editing; X.J. Data
- Management, Visualization; H.L. Data Management, Methodology; P.Z. Formal Analysis, Data
- Management; H.L. Writing Review & Editing, Funding Acquisition, Conceptualization, Supervision.

### 326 Competing interest

- The authors declare that they have no known competing financial interests or personal relationships that
- could have influenced the work reported in this paper.
- Acknowledgments. This research was supported by the National Key Research and Development
- Program of China (2021 YFD 1700900), the Industrial Technology System for Cultivated Land
- Protection in Henan Province (HARS-22-19-S), the Natural Science Foundation of Henan Province
- (Grant No. 252300420043), and the Key Research and Development Program of Henan Province (Grant
- No. 251111112200).

## Data availability

- All data are available in the text, Supplement or publicly on Zenodo (DOI 10.5281/zenodo.17639507).

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

# https://doi.org/10.5194/egusphere-2025-4460 Preprint. Discussion started: 18 November 2025 © Author(s) 2025. CC BY 4.0 License.

- X., Guo, Z., Chen, Y., Feng, L., He, S., Zhang, X., Lau, A. K. H., Tao, S., and Houlton, B. Z.: Fertilizer
- management for global ammonia emission reduction, Nature, 626, 792-798,
- https://doi.org/10.1038/s41586-024-07020-z, 2024.
- Yang, F., Tan, J., Zhao, Q., Du, Z., He, K., Ma, Y., Duan, F., Chen, G., and Zhao, Q.: Characteristics of
- PM<sub>2.5</sub> speciation in representative megacities and across china, Atmospheric Chem. Phys., 11, 5207-
- 5219, https://doi.org/10.5194/acp-11-5207-2011, 2011.
- Zhang, H., Hong, Z., Wei, L., Thornton, B., Hong, Y., Chen, J., and Zhang, X.: Stable isotopes unravel
- the sources and transport of PM2.5 in the Yangtze River delta, china, Atmosphere, 14,
- https://doi.org/10.3390/atmos14071120, 2023.
- Zhang, L., Altabet, M. A., Wu, T., and Hadas, O.: Sensitive measurement of  $NH_4^+$  15N/14N ( $\delta^{15}NH_4^+$ ) at
- natural abundance levels in fresh and saltwaters, Anal. Chem., 79, 5297-5303,
- https://doi.org/10.1021/ac070106d, 2007.
- Zhou, Y., Zheng, N., Luo, L., Zhao, J., Qu, L., Guan, H., Xiao, H., Zhang, Z., Tian, J., and Xiao, H.:
- Biomass burning related ammonia emissions promoted a self-amplifying loop in the urban environment
- in kunming (SW china), Atmos. Environ., 253, https://doi.org/10.1016/j.atmosenv.2020.118138, 2021.
- Zou, D., Sun, Q., Liu, J., Xu, C., and Song, S.: Seasonal source analysis of nitrogen and carbon aerosols
- of PM2.5 in typical cities of zhejiang, china, Chemosphere, 303,
- https://doi.org/10.1016/j.chemosphere.2022.135026, 2022.
