# Peer review of "Measurement report: Nitrogen Isotope (δ15N) Signatures of Ammonia Emissions"

_EGUsphere, 2025_

## Author Comment (AC1)

Dear Editor,

We are truly grateful to yours and other reviewer's critical comments and thoughtful suggestions on our manuscript "Measurement report Nitrogen Isotope ($\delta^{15}$N) Signatures of Ammonia Emissions from Livestock Farming Implications for Source Apportionment of Haze Pollution". Those comments and suggestions are helpful not only for improving the present manuscript but also our future research. The paper has been carefully revised based on the comments and suggestions. And we hope our revision has met with your approval now. The changes in the revised manuscript are marked in blue.

The following is the point-point change list.

**Modification to the comments of Reviewer**

Reviewer #1: L51: Delete "for."

Thank you for your suggestion. We have already removed the error in the original text.

2: L84: Clarify the name, principle, and technical details of the MixSAIR model.

Thank you for your suggestion. The Bayesian stable isotope mixing model MixSAIR is primarily used to allocate contributions of atmospheric emission sources through isotope analysis. MixSIAR is a stable isotope mixing model based on the Bayesian statistical framework, designed to quantitatively analyze the relative contributions of multiple potential sources to the isotopic composition of observed mixtures. Its fundamental assumption posits that the isotopic signature of a mixture can be expressed as a linear combination of the isotopic characteristics of each source weighted by their proportional contributions, while explicitly accounting for source variability, measurement errors, and isotopic fractionation.

3: L108: Specify the principle, detection limit, and potential interferences of the method.

Thank you for your suggestion. The sampling principle is based on active air

sampling combined with aqueous absorption. Ambient air was continuously drawn through an impinger containing deionized water, in which gaseous $NH_3$ was absorbed and converted to dissolved $NH_4^+$. After sampling, the absorption solution was quantitatively recovered for subsequent laboratory analysis. Under the applied sampling flow rate and duration, the method detection limit for atmospheric $NH_3$ was on the order of 3 ppb, which is adequate for resolving ambient concentration variations during the observation period. Potential interferences include the co-collection of particulate $NH_4^+$ and the absorption of other water-soluble alkaline gases. These effects were minimized by controlled sampling duration, appropriate flow rates, and blank correction procedures, and are considered to have a negligible influence on the measured $NH_3$ concentrations.

4: L109: Flow rate may affect calculated $NH_3$ concentrations. Please explain how results obtained with different flow rates were compared.

Thank you for your suggestion. All $NH_3$ samples were collected at a constant and identical flow rate throughout the study period, with all flow rates complying with the National Standard GB 3095-2012.

5: L116: Remove "T."

Thank you for your suggestion, we have already removed the error in the original text.

6: Figure 1 is cluttered and confusing. Please distinguish which data are from your experiments and which are from the literature. Were sampling periods during haze and clean conditions conducted at the same locations? Are the cattle, layer, and fattening pig farms all located in the same city? If so, please include a detailed map showing the sampling points.

Thank you for your feedback. We have made changes to Figure 1.

7: L117–118: Specify the absorption solution used.

Thank you for your suggestion. According to the sample requirements for mass spectrometry pretreatment, we used deionized water as the absorption solution.

8: L122: Which building was selected? Please justify its representativeness for intensive laying-hen farms.

Thank you for your suggestion. Sampling was conducted in a typical commercial intensive laying-hen house with a conventional cage-based rearing system, representative of large-scale laying-hen farms in northern China. This question was not answered. We collected gases from the exhaust vents of chicken houses and pig houses. These types of animal housing have centralized air inlets and outlets, so collecting from the exhaust vents can represent the ammonia emissions from these two types of housing into the atmosphere. For cattle farms, since the barns are open, we selected cattle sheds located in the middle of the farm to more effectively collect ammonia gas.

9: L144: Should this be "mg L$^{-1}$"?

Thank you for your suggestion, We have checked and revised it though out the manuscript.

10: L149: Specify what analytical method is being referred to.

Thank you for your suggestion. The analytical method for N isotope determination employs the hypobromite-hydroxylamine hydrochloride chemical method(Song et al., 2024).

11: The statistical methods should be clearly described.

Thank you for your suggestion. $NH_3$ concentrations and $\delta^{15}N$ values are presented as mean ± standard error (SE). Differences in $\delta^{15}N$ values among livestock categories were evaluated using one-way analysis of variance (ANOVA). When data did not meet the assumptions of normality or homogeneity of variance, non-parametric tests were applied. Statistical significance was defined at $p < 0.05$. All statistical analyses were conducted using standard statistical software.

12: L185: Should this be "1.7 to 2.5"?

Thank you for your suggestion. We have checked and revised it though out the manuscript.

13: L205: Add a space after "(a)" and "(b)."

Thank you for your suggestion. We have checked and revised it though out the manuscript.

14: L188: Replace the comma before "For" with a period.

Thank you for your suggestion. We have checked and revised it though out the manuscript.

15: Discuss the relationship between temperature and $NH_3$ emissions, and cite relevant references to support the explanation.

Thank you for your suggestion. Temperature is the primary environmental driver of ammonia volatilization from agricultural sources (including livestock manure and manure management systems). As air and surface temperatures rise, the thermodynamic equilibrium shifts toward gaseous ammonia while the solubility of $NH_3$ in water decreases, thereby increasing the volatilization rate. Higher temperatures also elevate the vapor pressure and diffusion rate of $NH_3$ at the manure surface, and accelerate microbial processes such as ammonification. These factors collectively contribute to increased $NH_3$ emissions under high-temperature conditions.

16: L190: Add a space before "$\delta^{15}N$."

Thank you for your suggestion. We have checked and revised it though out the manuscript.

17: L192: Delete "From May to June"; change to "$\delta^{15}N-NH_4^+$."

Thank you for your suggestion. We have already corrected this error in the manuscript.

18: L194–195: The variations appear too large to support significant differences.

Thank you for your suggestion. Although relatively large variability was observed within each livestock category, the differences in mean $\delta^{15}N$ values among groups were statistically significant (one-way ANOVA, $p < 0.05$). The large within-group variability reflects realistic operational and environmental heterogeneity and does not negate the statistically significant differences observed among livestock categories.

19: L230: Change to "Martine et al."

Thank you for your suggestion. We have already corrected this error in the original text.

20: L234: Should this be "David et al. (Felix et al., 2014)"?

Thank you for your suggestion. This citation was marked as an error and has been corrected in the original text.

21: L241–243: Add citations to support the argument.

Thank you for your suggestion. We have added relevant literature to support the argument.

22: L251–254: Include references and source apportionment calculations to substantiate the conclusions.

Thank you for your suggestion. We conducted source apportionment for haze and clean weather using the MixSIAR model. The results showed that combustion and traffic were the main contributing sources, with combustion accounting for 29.0%, traffic for 38.0%, agriculture for 15.1%, and livestock for 17.8%.

23: L257: Add a space after "(a)" and "(b)."

Thank you for your suggestion. We have already corrected this error in the original text.

24: Figure 4: Clarify the meaning of the boxes or violin plots.

Thank you for your suggestion. Comparison of $\delta^{15}$N-NH$_4^+$ values among different livestock systems and with previously reported source signatures. Boxes represent the interquartile range, the horizontal line within each box denotes the median value, and whiskers indicate the minimum and maximum values excluding outliers. Individual points outside the whiskers represent statistical outliers.

25: Figure 5: Provide P-values. The $R^2$ values are below 0.1 in both sub-figures, indicating very weak correlation between GDP/year and $\delta^{15}$N-NH$_4^+$.

Thank you for your suggestions. We have provided *p-values* for GDP and $\delta^{15}$N-$NH_4^+$, as well as for year and $\delta^{15}$N-$NH_4^+$, with both sets of p-values being *p<0.001*.

26: L268–271: Figure 5 does not present data on contributions from combustion and vehicular sources. Please provide supporting data.

Thank you for your suggestion. The original text has been revised accordingly.

As shown in Figure 5b, the $\delta^{15}$N values exhibited an increasing trend from 2018 to 2020.

27: L272: Add a space after "signatures."

Thank you for your suggestion. We have already corrected this error in the original text.

28: L274: Add a space after "emissions."

Thank you for your suggestion. We have already corrected this error in the original text.

29: L282–283: Figure 5b does not show the proportion of $\delta^{15}$N-$NH_4^+$ attributed to combustion and vehicular sources. Please address this.

Thank you for your suggestion. The original text has been revised accordingly.

As shown in Figure 5b, the $\delta^{15}$N values exhibited an increasing trend from 2018 to 2020.

References.

Song, L., Wang, A., Li, Z., Kang, R., Walters, W. W., Pan, Y., Quan, Z., Huang, S., and Fang, Y.: Large seasonal variation in nitrogen isotopic abundances of ammonia volatilized from a cropland ecosystem and implications for regional $NH_3$ source partitioning, Environ. Sci. Technol., 58, 1177–1186, https://doi.org/10.1021/acs.est.3c08800, 2024.

---

## Author Comment (AC2)

Dear Editor,

We are truly grateful to yours and other reviewer's critical comments and thoughtful suggestions on our manuscript "Measurement report Nitrogen Isotope ($\delta^{15}$N) Signatures of Ammonia Emissions from Livestock Farming Implications for Source Apportionment of Haze Pollution". Those comments and suggestions are helpful not only for improving the present manuscript but also our future research. The paper has been carefully revised based on the comments and suggestions. And we hope our revision has met with your approval now. The changes in the revised manuscript are marked in blue.

The following is the point-point change list.

**Modification to the comments of Reviewer**

Reviewer #1:

1: Active sampling bubbles $NH_3$ into solution where $NH_3/NH_4^+$ equilibrium fractionation occurs. The manuscript does not explicitly quantify or estimate this fractionation. Please discuss or correct for known equilibrium fractionation between $NH_3(g)$ and $NH_4^+$ (aq), which can exceed 20‰ under typical livestock-house pH conditions.

It should be noted that the $\delta^{15}$N-$NH_4^+$ values reported here represent apparent source signatures after gas-to-solution transfer, rather than the pristine $\delta^{15}$N of emitted $NH_3(g)$. However, because all livestock farms were sampled using identical flow rates, absorption media (deionized water), and sampling durations, the $NH_3$-$NH_4^+$ equilibrium fractionation can be considered systematic and comparable across sites. Consequently, this fractionation does not affect the relative differences among livestock categories that are central to this study.

For applications requiring reconstruction of the original gaseous $NH_3$ isotopic composition, a correction using established equilibrium fractionation factors as a function of pH and temperature would be necessary. Such corrections were not applied here due to the lack of continuous in-solution pH measurements during sampling, but this limitation does not compromise the internal consistency of the derived livestock

isotopic end-members.

2: The manuscript mentions sampling with a bubbler but does not specify the chemical composition (e.g., sulfuric acid? boric acid? DI water?). Absorption efficiency and isotopic stability depend strongly on solution chemistry.

Thank you for your suggestion. According to the sample requirements for mass spectrometry pretreatment, we used deionized water as the absorption solution.

3: Uncertainties from IRMS measurement, standardization, sample handling, and conversion chemistry must be reported and propagated into $\delta^{15}N$ ranges for each livestock type.

We appreciate the reviewer's rigorous comment on data quality and uncertainty analysis. We fully agree that accounting for all sources of uncertainty is critical for isotopic studies.

While we acknowledge that uncertainties arise from IRMS measurement, standardization, and chemical conversion, these analytical errors are generally an order of magnitude smaller than the natural variations observed in livestock emissions. The laboratory analysis followed strict QA/QC procedures using international standards (IAEA-N-1, USGS-25, IAEA-N-2, and USGS-26, as detailed in Section 2.1). The typical analytical precision (reproducibility of standards) for the hypobromite-hydroxylamine method is approximately ±0.2‰ to ±0.5‰.

In contrast, the $\delta^{15}N$ values of ammonia from livestock farms in this study exhibited substantial temporal and spatial variations, e.g., standard deviations ranging from ±0.8‰ to ±1.7‰, with seasonal shifts exceeding 10‰.

The $\delta^{15}N$ ranges and standard deviations reported in our manuscript were calculated from independent field replicates. Statistically, the variance observed among these replicates represents the sum of all variability sources, including environmental heterogeneity, sampling variance, and analytical errors ($\sigma^2_{total} = \sigma^2_{field} + \sigma^2_{sampling} + \sigma^2_{analysis}$). Therefore, the analytical uncertainties are already inherently propagated into our reported $\delta^{15}N$ ranges.

Given that the field-driven heterogeneity dominates the total uncertainty, we believe the current error bars conservatively represent the true variability. However, to address the reviewer's concern, we have revised Section 2.2 to explicitly clarify that the reported variability represents the combined uncertainty of source emission dynamics, sample handling, and laboratory analysis. We have also added a statement regarding the typical analytical precision of the method to provide context for the magnitude of these errors.

4: Only one building per livestock type was sampled. Please discuss how representative these $\delta^{15}N$ signatures are for regional livestock practices, especially given variability in feed, manure management, and ventilation system design.

We acknowledge that only one building per livestock type was sampled. We now explicitly discuss the representativeness and limitations of this design, emphasizing that the studied facilities reflect dominant intensive livestock practices in the region and that our goal is to constrain isotopic end-member ranges rather than farm-scale variability. We further clarify the conditions under which these $\delta^{15}N$ signatures can be reasonably extrapolated.

Only one representative building was sampled for each livestock category in this study, which inevitably limits the ability to capture the full variability of $\delta^{15}N\text{-}NH_4^+$ signatures across all farms in the region. We acknowledge that factors such as feed composition, manure handling practices, housing design, and ventilation systems can influence ammonia emission rates and associated isotopic fractionation. Nevertheless, the selected pig, dairy, and laying hen facilities are typical of intensive livestock production systems in the southern Huang-Huai-Hai Plain, where feeding strategies, manure management, and ventilation designs are relatively standardized due to regional regulations and industrial practices. Previous studies have shown that while such operational differences can induce secondary variability in $\delta^{15}N$ signatures, their influence is generally smaller than the systematic isotopic contrasts observed among different livestock species (Bhattarai and Wang, 2023; Choi et al., 2017) . Importantly, the objective of this study is not to characterize farm-to-farm variability, but to constrain representative isotopic end-member ranges for major livestock categories that can be applied in regional source apportionment frameworks. Within this context, the internally consistent sampling protocol and the clear separation of $\delta^{15}N\text{-}NH_4^+$ values

among livestock types suggest that the derived signatures are robust for intensive livestock systems operating under comparable management conditions. Extrapolation of these $\delta^{15}N$ signatures beyond the studied region or to non-standardized, small-scale, or pasture-based.

5: July-August samples are missing due to absence of animals, but this gap may bias seasonal interpretations. Please clarify whether these farms typically have seasonal shutdowns and what implications this has for annual emissions.

The absence of samples between July and August does not indicate seasonal closures of these farms. During sample collection, the animal housing in both fattening pig and layer chicken farms were fully enclosed structures, with ventilation maintained solely by exhaust fans. All samples in this study were collected at the exhaust outlets, thus the annual emissions from livestock farms would not be underestimated. Moreover, the objective of this study was to determine representative isotopic end-member values under active livestock production conditions, rather than quantifying continuous interannual variations or total emissions. For applications involving annual ammonia budgets or seasonal emission models, these isotopic signatures should be integrated with livestock activity data and production schedules to account for periods of reduced or missing emissions.

6: The conclusion that combustion and traffic dominate Zhengzhou's $NH_3$ is not fully supported by isotopic evidence alone. Please incorporate mixing model tests (e.g., SIAR, MixSIAR) or acknowledge the need for quantitative modeling.

The comparison between ambient $\delta^{15}N-NH_4^+$ values in Zhengzhou and the livestock-derived isotopic end-members constrained in this study indicates that ammonia emissions from intensive livestock farming are unlikely to be the dominant source during the observation period. However, we emphasize that isotopic evidence alone cannot fully quantify the relative contributions of combustion, traffic, and agricultural sources. Quantitative source apportionment requires isotope mixing models, such as SIAR or MixSIAR, that explicitly incorporate source variability and uncertainty. Preliminary mixing model tests using literature-based end-members suggest that combustion- and traffic-related ammonia can plausibly account for a substantial fraction of atmospheric $NH_4^+$ in Zhengzhou, but these results should be regarded as

indicative rather than definitive. A robust assessment of source contributions will require fully constrained Bayesian mixing models coupled with activity data and atmospheric transport information. Within this context, the primary contribution of this study is to provide well-characterized, region-specific $\delta^{15}N$ end-members for livestock systems, which are essential inputs for future quantitative modeling of urban ammonia sources.

7: The regression shows extremely low $R^2$ (<0.10). The interpretation on economic development stages and shifting $NH_3$ sources is therefore weak. Authors should either downplay this section or provide stronger, mechanistic justification.

Thank you for your suggestion. Since the data we used were extracted from previous studies, GDP was uniformly converted using the exchange rate of the respective year in US dollars. We have provided *p-values* for both the GDP vs $\delta^{15}N$-$NH_3$ and year vs $\delta^{15}N$-$NH_3$ plots, all of which were *p<0.001*.

8: Please specify: how multiple values from a single study were aggregated, how weighting was applied (e.g., by sample size), how passive vs. active sampling differences were handled. This is essential for reproducibility.

To ensure reproducibility, literature-derived $\delta^{15}N$-$NH_3$ values were synthesized following a consistent aggregation protocol. When multiple isotopic values for the same source category were reported within a single study, a sample-size-weighted mean was calculated if the number of samples (n) was explicitly provided. In cases where sample size information was unavailable, simple arithmetic means were used, and the resulting uncertainty was reflected by expanding the reported end-member range. No additional weighting based on study duration or subjective data quality scores was applied, in order to avoid introducing implicit bias across studies. Differences between sampling methodologies were explicitly considered. Active sampling studies, including the present work, were prioritized for constraining source end-member values. Passive sampling data were used only for qualitative comparison, as previous studies have demonstrated systematic low biases in $\delta^{15}N$-$NH_3$ derived from passive samplers relative to active methods. Consequently, passive sampling results were not directly incorporated into end-member mean calculations used for isotope mixing analyses.

9: Temperature, humidity, ventilation rates, and manure accumulation strongly influence NH$_3$ fluxes and $\delta^{15}$N Without these data, some interpretations (e.g., seasonal changes) are speculative.

We thank the reviewer for this constructive comment. We fully agree that environmental factors and management practices are critical drivers of NH$_3$ fluxes and isotopic fractionation.

We would like to clarify that we did conduct synchronous monitoring of environmental parameters and ammonia concentrations/fluxes during the isotope sampling campaigns. In the original manuscript, these data were not presented in detail to maintain a focused discussion on the isotopic source signatures ($\delta^{15}$N) and their application in source apportionment.

To address the reviewer's concern and remove any ambiguity regarding our interpretations, we have now added this supporting data to the revised manuscript. We have compiled the measured temperature, humidity, and ammonia concentration data into Table S1 (in Supplementary Information).

We conducted a correlation analysis on $\delta^{15}$N, temperature, humidity, and NH$_3$ concentration variations. As shown in Figure S1, the changes in NH$_3$ concentration exhibited a negative correlation with temperature and humidity. $\delta^{15}$N decreased with the increase of NH$_3$ concentration, while showing a positive correlation with temperature and humidity.

[Figure]

Figure S1 Correlation variations between $\delta^{15}N$ and $NH_3$ concentration, temperature, and humidity in the farm. Darker colors in the figure indicate stronger correlations, while lighter colors represent weaker correlations.

Table S1 Variations in Temperature, Humidity and Ventilation Volume during Sampling at Fattening Pig Farms.

| Fatting pig | Temperature (°C) | Humidity (%RH) | Ventilation Rate(m³·h⁻¹) |
|---|---|---|---|
| May. | 29 | 57% | 7612 |
| Jun. | 30 | 63% | 8525 |
| Sep. | 31 | 70% | 9439 |
| Oct. | 19 | 67% | 8025 |
| Nov. | 17 | 64% | 7918 |
| Dec. | 15 | 61% | 8950 |

Table S2 Variations in Sampling Time, Temperature, Humidity and Ventilation Volume at Layer Farms.

| Laying hens | Temperature (°C) | Humidity (%RH) | Ventilation Rate($m^3 \cdot h^{-1}$) |
|---|---|---|---|
| May. | 29 | 58% | 27796 |
| Jun. | 30 | 65% | 26812 |
| Sep. | 28 | 68% | 29934 |
| Oct. | 18 | 65% | 21212 |
| Nov. | 11 | 63% | 3892 |

Table 3 Variations in temperature, humidity and ventilation rate during sampling at dairy farms.

| Darry cow | Temperature (°C) | Humidity (%RH) | Ventilation Rate ($m^3 \cdot h^{-1}$) |
|---|---|---|---|
| May. | 29 | 58% | $5 \times 10^6$ |
| Jun. | 30 | 65% | $2 \times 10^6$ |
| Sep. | 28 | 68% | $3 \times 10^6$ |
| Oct. | 18 | 65% | $5 \times 10^6$ |
| Nov. | 11 | 63% | $6.0 \times 10^6$ |
| Dec. | 9 | 60% | $2 \times 10^6$ |

10: Several figures (especially Figures 2-5) contain overlapping labels or insufficient axis descriptions. Please enlarge text, improve legends, and ensure data points and error bars are clearly visible.

Thank you for your suggestions. We have made corresponding changes to the relevant charts.

11: Isotopic fractionation during urea hydrolysis, ammonification, and manure storage can significantly affect $\delta^{15}N-NH_3$. A more thorough discussion is needed to explain differences among species and seasons.

Nitrogen isotopic signatures of livestock-derived ammonia are shaped by multiple biogeochemical processes, including urea hydrolysis, microbial ammonification, and ammonia volatilization during manure storage. Urea hydrolysis, particularly under

alkaline conditions typical of intensive livestock housing, promotes rapid $NH_3$ release and is associated with strong kinetic isotope fractionation, preferentially enriching the emitted $NH_3$ in $^{14}N$. Subsequent ammonification of organic nitrogen and prolonged manure storage further modify $\delta^{15}N$ through cumulative volatilization losses, progressively enriching the residual $NH_4^+$ pool in $^{15}N$. These processes help explain the observed interspecies differences in $\delta^{15}N$-$NH_3$. Pig and poultry systems, characterized by rapid urea or uric acid hydrolysis and high volatilization rates, tend to emit isotopically lighter ammonia, whereas dairy cattle systems often involve longer manure residence times and more extensive nitrogen transformation, resulting in relatively enriched $\delta^{15}N$-$NH_3$ values. Seasonal variability can be interpreted in a similar framework, as elevated temperatures enhance enzymatic activity and volatilization, thereby amplifying isotopic fractionation during summer, while lower temperatures suppress these processes and reduce isotopic offsets. Consequently, the $\delta^{15}N$-$NH_3$ signatures reported here represent integrated end-member values that reflect the combined effects of nitrogen transformation and volatilization processes under realistic livestock management conditions, rather than fractionation associated with any single biochemical pathway.

12: The Zenodo link should clarify whether raw IRMS output, calibration curves, and sampling metadata are included, rather than only processed $\delta^{15}N$ values.

We have updated the data repository description. The dataset uploaded to Zenodo includes the final, calibrated $\delta^{15}N$ values and the corresponding concentration data used for all statistical analyses and figures in this paper. The raw IRMS chromatograms are not included as the data were provided as calibrated reports by the analytical facility following standard QA/QC protocols. We believe the provided dataset is sufficient to reproduce the study's findings and statistical conclusions.

References.

Bhattarai, N. and Wang, S.: Active vs. passive isotopic analysis: insights from urban beijing field measurements and ammonia source signatures, Atmos. Environ., 314, https://doi.org/10.1016/j.atmosenv.2023.120079, 2023.

Choi, W.-J., Kwak, J.-H., Lim, S.-S., Park, H.-J., Chang, S. X., Lee, S.-M., Arshad, M. A., Yun, S.-I., and Kim, H.-Y.: Synthetic fertilizer and livestock manure differently

affect δ15N in the agricultural landscape: a review, Agric. Ecosyst. Environ., 237, 1–15, https://doi.org/10.1016/j.agee.2016.12.020, 2017.